# RATING CONTINUOUS ACTIONS IN SPATIAL MULTI-AGENT PROBLEMS

## ABSTRACT

We study credit assignment problems in spatial multi-agent environments where agents pursue a joint objective. On the example of soccer, we rate the movements of individual players with respect to their potential for staging a successful attack. We propose a purely data-driven approach to simultaneously learn a model of agent movements as well as their ratings via an agent-centric deep reinforcement learning framework. Our model allows for efficient learning and sampling of ratings in the continuous action space. We empirically observe on historic soccer data that the model accurately rates agent movements w.r.t. their relative contribution to the collective goal.

## 1 INTRODUCTION

Learning to assess individual behavior in an environment that contains several agents with potentially concurring goals received a lot of attention recently (Lowe et al., 2017; Foerster et al., 2017), spurred in part by the large success of deep reinforcement learning models in single agent domains.

In this paper, we focus on cooperative multi-agent problems in spatial environments on the example of soccer. Figure 1 illustrates a snapshot of a professional soccer game where individual players behave such that the collective is likely to score. We aim to assign a rating to every agent's movement that reflects the quality w.r.t. the collective goal. To do so, we propose to learn possible (and likely) movements of individual players in the unconstrained continuous space.

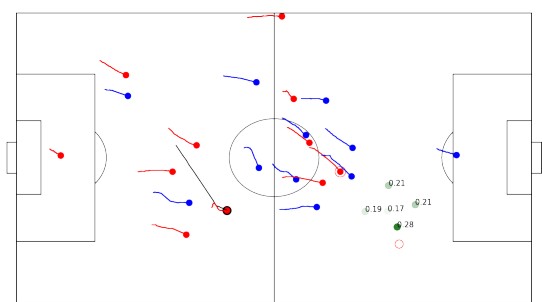

We translate the spatial multi-agent problem into a Markov Decision Process (MDP) where agents perceive positions and movements of other agents as a state representation that is possibly augmented by environmental variables such as the position of the ball. Given

Figure 1: Learned ratings of potential movements of the circled red player (green dots, the darker the higher the rating). The red circle denotes the true movement that the player is about to perform.

the state, every agent decides on a movement from a continuous action space that aims to support the team in receiving a joint reward. Successful strategies that ground on simulation (Copete et al., 2015; Hausknecht & Stone, 2016) are unfortunately not applicable in our scenario where historic games are the only data source. On the other hand, credit assignments by experts are tedious and only available for a few manually selected scenes per game and usually ground on heuristics (Link et al., 2016; Fernandez & Bornn, 2018). To remedy the need of experts, we take a purely data-driven approach without external information and focus on learning from observations. On the example of soccer, we show that our complementary approach to expert-based ratings gives meaningful scores that highly correlate with manually created ones by experts.

Technically, we propose an agent-centric deep reinforcement learning approach to rate agent movements in spatio-temporal decision making problems. The real distribution governing each agent's movements in the continuous action space is unknown and can only be observed from examples. We

introduce a mixture model that models the unknown distribution of actions, allowing for efficient learning, sampling and evaluation of movements when the examples are limited.

Given the mixture model, we devise a deep reinforcement learning-based approach to learn the values of the mixture components (the ratings of actions). Our approach allows to efficiently evaluate the advantage function of actual movements of the agents in any situation. Both, policy and ratings of movements, are learned simultaneously by jointly optimizing the likelihood and a variant of the Bellman error. The deep model renders the input independent from the roles and positions of the agents and convolutions capture the entire environment from each agent's perspective. Empirically, we show that our approach leads to sensible ratings which are comparable with expert grades on professional soccer data.

## 2 RELATED WORK

Cooperative multi-agent scenarios where multiple agents act in a shared environment and pursue a joint objective are usually addressed with multi-agent RL techniques (Tan, 1993; Littman, 2001). These approaches are often attributed to either *team learning* aiming at models in joint action spaces or *concurrent learning* focusing on individual agents (Panait & Luke, 2005). Claus & Boutilier (1998) demonstrate that the convergence of cooperative concurrent learning could be nearly obtained as in a single-agent setting. Omidshafiei et al. (2017) present a deep learning approach for partially observable multi-agent tasks. Deep RL has been successfully applied in game contexts (Mnih et al., 2015; Lanctot et al., 2017; Silver et al., 2016; 2017). Moreover, Foerster et al. (2017) focus on learning decentralized policies while a central critic is optimizing independent behaviors.

An actor-critic deep model for an abstraction of RoboCup has been proposed to learn the value of actions (Hausknecht & Stone, 2016). The action space contains four discrete actions with continuous parameters, one of which captures the magnitude and direction of movement for each agent and thus resembles our movement policies. Agents learn to play the game from scratch in a fully simulated setting, rendering the approach unsuited for problems like team sport that cannot (yet) be simulated.

Zhan et al. (2018) and Le et al. (2017b) employ deep imitation learning to model coordinated movements of agents in the problem of basketball and soccer, respectively. Similarly, Le et al. (2017a) use imitation learning to compare the behavior of soccer players to the average performance of either the player herself or a team average. Although the analysis is based on very interesting movement patterns, the average performance does not serve as a reliable evaluation of player movements as it is oblivious of situations and contexts. By contrast, our approach evaluates whether an agent's movement leads to maximizing the actual reward of an episode.

Player rating in soccer is a fairly new topic which aims at quantitatively ranking players instead of subjective measures. The related works consist of defining game indexes (McHale et al., 2012), space generated by players (Fernandez & Bornn, 2018), and the location of completed passes (Brooks et al., 2016). We present a model to learn the value of players movements that gives a fine grained measure of players performance. Other computational approaches of analyzing soccer matches often rely on domain knowledge (Lucey et al., 2014; Bialkowski et al., 2014) or deploy frequency-related criteria to narrow down the candidate space (Haase & Brefeld, 2014). Scoring opportunities are for instance rated by Spearman (2018) and Link et al. (2016).

## 3 LEARNING TO RATE ACTIONS

### 3.1 PRELIMINARIES

Markov decision processes consist of a state space $S$, a state-dependent action space $M(s), s \in S$, a reward function $r(s, m, s^+)$, where $s, s^+ \in S$, and $m \in M(s)$, and a state transition distribution $p(s^+|s, m)$. We denote the next state and next action by $s^+$ and $m^+$, respectively. A stochastic policy $\pi(m|s)$ describes the distribution of actions $m \in M(s)$ that agents in state $s$ will decide for.

In our multi-agent setting, state and action spaces are jointly defined for a set of $n$ agents. A state $s \in S$ describes the positions of the agents, $\{f^1, ..., f^n\}$, augmented by some other features $\{g^1, ..., g^n\}$ such as their direction of movement or team ID. Additionally, $s$ encodes the environmental features, e.g., the position and speed of the ball, as well as the location of the goals. An action $m \in M(s)$ is

defined by the set of movements $m = \{m^1, ..., m^n\}$ of the agents between two consecutive states $s$ and $s^+$, resulting in a continuous action space.

A soccer game is split into episodes of uninterrupted ball possession phases of a team, hence rendering episodes of variable length. A reward is granted at the last state transition of every episode. We assume a positive reward if the attacking team carries the ball into the dangerous zone of the opponent while retaining ball possession, and zero otherwise. However, note that any other definition of a positive reward may be used as well.

State transition probabilities, $p(s^+|s, m)$, are partly deterministic since an agent's position is the sum of previous positions and the performed movements, $f^{+i} = f^i + m^i$. Nevertheless, the movement of the ball is a part of the environment and is modeled as random movement. Therefore, the movement distribution is governed by a stochastic policy $\pi(m|s)$. Although the quantity is unknown, samples from $\pi(m|s)$ are contained in the episodes. We compute the ratings of the movements using the Bellman equation of the form

$$Q(s, m) = \mathbb{E}_{s^+ \sim p}\Big[r(s, m, s^+) + \gamma \mathbb{E}_{m^+ \sim \pi(\cdot|s^+)} Q(s^+, m^+)\Big]. \tag{1}$$

The value function recursively computes the cumulative discounted reward until the end of an episode where $\gamma \in (0, 1]$ is a discount factor. In the remainder, we use $r$ instead of $r(s, m, s^+)$ for simplicity in the notation. Equation 1 gives the value of the collective movements in the current state, while we are interested in valuating the movements of individual agents $m^i$. Hence, we define

$$\bar{Q}(s, m^i) = \int_{\tilde{m}:\tilde{m}^i = m^i} \frac{\pi(\tilde{m}|s)}{\bar{\pi}(m^i|s)} Q(s, \tilde{m}) d\tilde{m} = \int_{\tilde{m}:\tilde{m}^i = m^i} p(\tilde{m}|s, m^i) Q(s, \tilde{m}) d\tilde{m} \tag{2}$$

as the value of movement $m^i$ of agent $i$ in state $s$, where $\bar{\pi}(m^i|s) = \int_{\tilde{m}:\tilde{m}^i = m^i} \pi(\tilde{m}|s) d\tilde{m}$. In other words, the value of a single agent's movement is the expectation over all collective movements in which that agent performs that movement.

Finally, when rating an agent's behavior, we generally focus on the *relative* value of a movement compared to the values of alternative movements the agent could perform in the same state. The relative gain of an action is often called the *advantage* of that action and is defined as

$$A(s, m^i) = \bar{Q}(s, m^i) - \int_{\tilde{m}^i} \pi(\tilde{m}^i|s) \bar{Q}(s, \tilde{m}^i) d\tilde{m}^i. \tag{3}$$

Note that the integral on the right side, that is, the state value function $V(s) = \int_{\tilde{m}^i} \pi(\tilde{m}^i|s) \bar{Q}(s, \tilde{m}^i) d\tilde{m}^i$, is the same for all agents.

### 3.2 THE OBJECTIVE FUNCTION

A general approach to learning the value function $\bar{Q}$ is to interpret the Bellman equation (Equation 1) as an optimality criterion by minimizing the *temporal differences*. We only have access to a finite amount of historic data consisting of episodes of state, movement, and reward, i.e., $(s, m, r) \in \{(s_t, m_t, r_t)\}_{t=1}^{t_e}$, where $t_e$ is the length of an episode. The Bellman objective is given by $\Omega_{\bar{Q}}$,

$$\Omega_{\bar{Q}} = \sum_{(s,m,r,s^+)} \sum_i \left(\bar{Q}(s, m^i) - \left(r + \gamma \int_{\tilde{m}^{+i}} \pi(\tilde{m}^{+i}|s^+) \bar{Q}(s^+, \tilde{m}^{+i}) d\tilde{m}^{+i}\right)\right)^2, \tag{4}$$

where every $(s, m^i)$ is a sample of the policy $\pi$ and thus $\Omega_{\bar{Q}}$ is an objective for an on-policy policy evaluation from a fixed data set.

A standard approach to learning $\bar{Q}$ is using a purely sample based method which approximates the integral with a single observed sample which leads to the following loss function

$$\Omega_{\bar{Q}_B} = \sum_{(s,m,r,s^+,m^+)} \sum_i \left(\bar{Q}(s, m^i) - \left(r + \gamma \bar{Q}(s^+, m^{+i})\right)\right)^2. \tag{5}$$

However, the number of training examples is limited compared to the complexity of the continuous action space for optimizing such objective. Additionally, we are interested in sampling realistic

movements that a player can actually perform. For instance, while in theory the value of running toward goal with a speed of 45 km/h is very high, practically, no player will be able to achieve it. Hence, in order to find the best possible movement of a player in a given state, the model should only be queried for possible actions. This problem arises since the continuous action space is essentially unconstrained and without additional information on the boundaries of player's physical capabilities. Even with this information, enforcing those bounds can be hard as investigated by Hausknecht & Stone (2016) who proposed to adjust gradients accordingly during learning. Another approach is to discretize the space into a predefined grid (Zheng et al., 2016).

Instead, we propose a model-based approach to learn the movement policy of the agents that imposes no constraints on the space of movements a priori. In the following, we introduce our model-based approach by devising parametric versions of $\bar{Q}(s, m^i)$ and $\pi(m^i|s)$ that are simultaneously adapted to data by a deep architecture to minimize $\Omega_{\bar{Q}}$ of Equation 4. Note that the integral over all subsequent movements $m^{+i}$ depends on the form of $\pi$ and might be difficult to compute. Nonetheless, our parametric approach allows to replace the integral by a computationally inexpensive sum which leads to efficient computation of $Q$, $A$, and $V$ (see Section 3.4).

The next section presents a general model for $\pi$ together with an optimization criterion to learn $\pi$ from the data. Section 3.4 then introduces a model for $\bar{Q}$ that makes use of $\pi$ and leads to a second optimization criterion. Finally, Sections 3.5 and 3.6 bring the loose ends together and present a deep architecture that simultaneously optimizes both criteria.

## 3.3 ESTIMATING POLICY $\pi$

As described in sections 3.1 and 3.2, the policy $\pi(m^i|s)$ should allow for both computationally efficient integration in Equation 4 and efficient sampling of *possible* movements for $m^i$. We therefore model $\pi$ as a mixture of $k$ two-dimensional Gaussians

$$\pi(m^i|s) = \sum_{l=1}^{k} \hat{\pi}_{\Psi_{\hat{\pi}}}(l|s, i) \mathcal{N}(m^i | \hat{m}_{\Psi_{\hat{m}^l}}^l(s, i), \sigma_{\Psi_{\sigma^l}}^l(s, i) \cdot \mathbb{I}), \tag{6}$$

where the two-dimensional function $\hat{m}_{\Psi_{\hat{m}^l}}^l(s, i)$ computes the mean of the $l$-th Gaussian of agent $i$ in state $s$. Although parameters $\Psi_{\hat{m}^l}$ are shared across agents, the means of the mixture model are computed for each player $i$ individually. The model allows to sample values efficiently at those $k$ means (see also Figure 1 for an example with $k = 5$).

Similarly, function $\sigma_{\Psi_{\sigma^l}}^l(s, i)$ maps state $s$ to the width parameter of the spherical covariance matrix $\sigma_{\Psi_{\sigma^l}}^l(s, i) \cdot \mathbb{I}$ of the $l$-th Gaussian of agent $i$. $\mathbb{I}$ denotes the two-dimensional identity matrix. Again, parameters $\Psi_{\sigma^l}$ are shared among all agents. The probability $\hat{\pi}_{\Psi_{\hat{\pi}}}(l|s, i)$ that agent $i$ chooses action $l$ in state $s$ is parameterized by $\Psi_{\hat{\pi}}$. The structure of $\Psi_{\hat{\pi}}$, $\Psi_{\hat{m}^l}$, and $\Psi_{\hat{\sigma}^l}$ are defined by the deep model in Section 3.6. To not clutter notation unnecessarily, we omit the parameters in the remainder whenever possible.

We estimate the parameters of the Gaussian mixture model by minimizing the negative log-likelihood on the samples. For that we need to estimate $p(s^+|s)$. Let $s^M$ be the part of the state that is defined by movements of agents such that $s^{+M} = \{f^{+i}; \forall i\}$, where $f^{+i} = f^i + m^i$, and let $s^B$ denote the remaining part. Then it gives

$$p(s^+|s) = p(s^{+B}, s^{+M}|s) = p(s^{+B}|s^{+M}, s) p(s^{+M}|s).$$

Although $p(s^{+B}|s^{+M}, s)$ remains unknown, samples of $p(s^{+B}|s^{+M}, s)$ can be drawn from the data. Therefore, minimizing the negative log-likelihood of $p(s^+|s)$ over data samples $\{(s, s^+)\}$ reduces to $p(s^{+M}|s)$ instead. We further assume that movements of agents are independent given the state and rewrite the state transition function in terms of Equation 6 as $p(s^{+M}|s) = \prod_i \pi(m^i|s)$.

The negative log-likelihood of the movement policy, $\Omega_L(\Psi_{\hat{\pi}}, \Psi_{\hat{m}^l}, \Psi_{\hat{\sigma}^l})$, thus is written as

$$\Omega_L(\Psi_{\hat{\pi}}, \Psi_{\hat{m}^l}, \Psi_{\hat{\sigma}^l}) = -\sum_{(s,m,s^+),i} \log \left( \sum_l \hat{\pi}_{\Psi_{\hat{\pi}}}(l|s, i) \mathcal{N}(m^i | \hat{m}_{\Psi_{\hat{m}^l}}^l(s, i), \sigma_{\Psi_{\sigma^l}}^l(s, i)) \right). \tag{7}$$

### 3.4 LEARNING VALUE FUNCTION $\bar{Q}$

Recall that the discrete set of movements as defined by the means of the mixture model of agent $i$ in state $s$ is given by $\{\hat{m}^1(s,i), ..., \hat{m}^k(s,i)\}$. Let $\hat{Q}_{\Psi_{\hat{Q}}} : (s, \hat{m}^l(s,i)) \mapsto \Re$ be a value function that maps those state-movement pairs to a real number with parameters $\Psi_{\hat{Q}}$. In order to generalize to all possible movements, i.e., movements that are not contained in $\{\hat{m}^1(s,i), ..., \hat{m}^k(s,i)\}$, the general value function $\bar{Q}$ computes the expected value

$$\bar{Q}(s, m^i) = \sum_l p(\hat{m}^l(s,i)|s, m^i, \sigma^l(s,i))\hat{Q}(s, \hat{m}^l(s,i)),$$

where the expectation is w.r.t. $p(\hat{m}^l(s,i)|s, m^i, \sigma^l(s,i))$, the posterior for the movement being generated by the $l$-th component of the movement model. The following proposition shows that the integral in Equation 4 can be substituted with a sum over the mixture components.

**Proposition 1.** *The integral in Equation 4 can be written as*

$$\int_{\tilde{m}^{+i}} \pi(\tilde{m}^{+i}|s^+)\bar{Q}(s^+, \tilde{m}^{+i})\, d\tilde{m}^{+i} = \sum_l \hat{\pi}(l|s^+, i)\hat{Q}(s^+, \hat{m}^l(s^+, i)).$$

*Proof.* (see Appendix A) □

Combining this result with $\bar{Q}$ yields a simplified form of $\Omega_{\bar{Q}}$ that reduces the learning of $\bar{Q}$ to learning the parametric function $\hat{Q}$ which is given by

$$\Omega_{\bar{Q}}(\Psi_{\hat{Q}}; \Psi_{\hat{\pi}}, \Psi_{\hat{m}^l}, \Psi_{\hat{\sigma}^l}) = \sum_{(s,m,r,s^+),i} \Bigg[ \sum_l p(\hat{m}^l_{\Psi_{\hat{m}^l}}(s,i)|s, m^i, \sigma^l_{\Psi_{\hat{\sigma}^l}}(s,i))\hat{Q}_{\Psi_{\hat{Q}}}(s, \hat{m}^l_{\Psi_{\hat{m}^l}}(s,i)) \tag{8}$$
$$- \Bigg( r + \gamma \sum_{l'} \hat{\pi}_{\Psi_{\hat{\pi}}}(l'|s^+, i)\hat{Q}_{\Psi_{\hat{Q}}}(s^+, \hat{m}^{l'}_{\Psi_{\hat{m}^{l'}}}(s^+, i)) \Bigg) \Bigg]^2,$$

### 3.5 PUTTING THINGS TOGETHER

The optimization criteria for learning $\pi$ and $\bar{Q}$ are determined in the last two sections. Equation 4 states a model-based learning objective for learning a generic value function $\bar{Q}$ that depends on movement policy $\pi$, but the parametric functions for $\pi$ and $\bar{Q}$ was not defined at that point. We then specified $\pi$ using a Gaussian mixture model with an optimization criterion for adapting its parameters to data. In the last step, this $\pi$ was used to define a value function $\bar{Q}$ whose parameters are learned by minimizing $\Omega_{\bar{Q}}$ in Equation 8. Our choice of $\pi$ allows to replace the integral in Equation 4 by a sum which leads to an efficiently trainable optimization problem.

In addition, since the computation of $\bar{Q}$ depends on $\pi$, the optimality of $\bar{Q}$ too depends on the parameters of $\pi$ in the optimization process $\Omega_{\bar{Q}}(\Psi_{\hat{Q}}; \Psi_{\hat{\pi}}, \Psi_{\hat{m}^l}, \Psi_{\hat{\sigma}^l})$. Despite the fact that $\pi$ and $\bar{Q}$ could be learned in a pipe-lined approach one after another, we propose a deep model that allows to learn all parameters $\Psi = \Psi_{\hat{Q}} \cup \Psi_{\hat{\pi}} \cup \Psi_{\hat{m}^l} \cup \Psi_{\hat{\sigma}^l}$ in one single optimization problem

$$\min_{\Psi} \quad (\Omega_{\bar{Q}}(\Psi_{\hat{Q}}; \Psi_{\hat{\pi}}, \Psi_{\hat{m}}, \Psi_{\hat{\sigma}}) + \Omega_L(\Psi_{\hat{\pi}}, \Psi_{\hat{m}}, \Psi_{\hat{\sigma}})).$$

We benefit from the convolutional model which enables sharing large parts of the individual parameter sets, thus speeding up training as well as prediction.

### 3.6 DEEP MODEL

In this section, we present the deep convolutional model that is developed to learn both $\bar{Q}$ and $\pi$. More detailed information can be found in Appendix B.

**Input:** Recall that the state consists of the position and direction of movements of all agents and the ball, augmented by additional information. The input to the net is thus represented by a three dimensional 'image-like' tensor $\mathsf{I} \in \mathbb{R}^{105 \times 68 \times 12}$ shown in Figure 2.

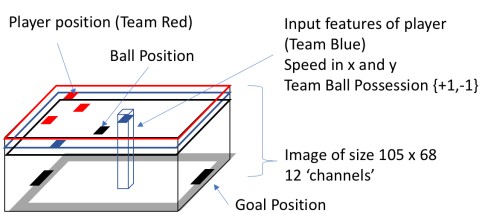

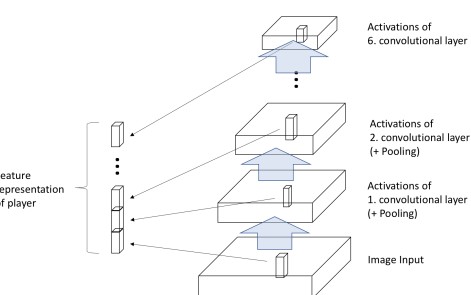

Figure 2: Input representation to the deep convolutional net. Layers: Team red, team blue, ball, speed in $x$-direction team red, speed in $y$-direction team red, speed in $x$ ball,..., team has ball possession $\{+1, -1\}$, goal position, border.

Figure 3: Feature representation for an agent as defined by the deep model. Activations of all convolutional layers at the position of the agent are concatenated to form feature representations that are fed into the loss net.

**Architecture:** The introduced tensor is the input to a 6-layer deep convolutional net such that the receptive field of activations at the last layer spans over the whole pitch. The exact definition of each layer can be found in Appendix B. Full feature representations of agents are computed by concatenating activations of convolutional layers as depicted in Figure 3.

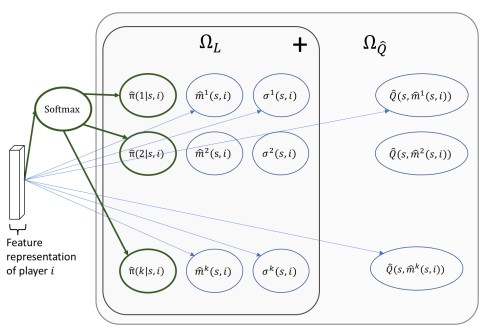

Figure 4: Output nodes and loss.

Hence, slices through all activation tensors are computed relative to the agent's position. For instance, if the size of the output tensor of the $j$-th convolutional layer is $n_j \times m_j \times o_j$ and the players position in the input image is $x, y$, the relevant slice of the input tensor is located at position $\lfloor x \cdot 105/n_j \rfloor, \lfloor y \cdot 68/m_j \rfloor$ and has length $o_j$. Slices through the outputs of all convolutional layers are computed and then compiled into a single feature vector that consists of features of different levels of granularity and receptive fields, centered at the agent's position. Representation vectors of agents have $d$ dimensions with $d = 12 + \sum_{j=1...6} o_j$.

An agent's feature representation connects to $5k$ outputs via a fully connected layer where $k$ is the number of components in the Gaussian mixture model defined in Equation 6 (cmp. Figure 4). Activations are linear for $\hat{m}$ and $\hat{Q}$, whereas $\hat{\pi}$ uses an additional softmax layer to ensure valid probability distributions. Only $\sigma$ has an exponential activation function to ensure positiveness.

## 4 EMPIRICAL STUDY

We evaluate the performance of our model on trajectory data from the UEFA Women's Euro 2017 tournament. The dataset consists of six games of the Dutch team from group stage to the final match. Each game comes as a sequence of $x$ and $y$-coordinates of all players and the ball sampled at 10 frames per second. Every frame is turned into an input tensor as described in Figure 2.

We extract all episodes of open play were one team continuously retains ball possession without the game being halted. An episode is considered successful if the team carries the ball into an area that extends 2 meters outside the penalty box of the opponent. In case of a successful action, the episode is labeled with a positive reward, otherwise the reward is 0. We report on results of leave-one-game-out cross validations; that is, every result is the mean of a 6-fold cross validation where each fold contains all episodes of exactly one of the six games. Error bars indicate standard error.

### 4.1 BASELINES

As mentioned in Section 3.2, a common approach to learn the value function from data is to approximate the integral in Equation 4 with a single observed sample, yielding the objective in Equation 5.

We compare our model to this baseline. However, as mentioned in Section 3.1, when rating players' movements, we are generally interested in the *advantage* function of a movement to assess its quality. In order to compute the advantage values, we need an estimate of the state value function $V$ (cmp. Equation 3). We consider three approaches to estimate $V$. The first two methods estimate the state value function from the learned $\bar{Q}_B$, i.e., we approximate the integral $V(s) = \int_m \pi(m|s)\bar{Q}_B(s,m)dm$ with a sum over $J$ sampled movements $V(s) \approx 1/J \sum_{j=1}^{J} \bar{Q}_B(s,m_j)$. Since the real movement policy $\pi$ is unknown, we uniformly sample from the space of possible movements[1], which forms the first baseline called *Baseline with Random Sampling*. The second baseline uses the policy $\bar{\pi}$ from our model to sample from the space of movements and is called *Baseline with Model*.

The third baseline directly estimates $V(s)$ using the same deep architecture as defined in Section 3.6. Instead of predicting values based on an agent's feature representation, a fully connected layer with a single output $V(s)$ is connected to the last convolutional layer. Thus, $V$ and $Q$ share all model parameters except those of the output nodes. $V$ is estimated by minimizing the expected temporal difference error for states $\min_V \sum_{(s,s^+,r)} (V(s) - (r + \gamma V(s^+)))^2$. We show that the learned $V$ assigns meaningful values to states in the Appendix C.1.

## 4.2 Movement Valuation

The ability to rate individual movements allows us to determine how good a player behaves on the soccer pitch. An example of a rating for one player is shown in Figure 1 for $k = 5$ mixture components. The hypothesis is that the more agents choose good actions, the more successful the team coordination and thus the higher the reward. [2] We validate the hypothesis as follows.

**Evaluation Measures:** We present metrics for evaluating both $Q$-values as well as advantage function $A$. To evaluate $Q$, we compute the $Q$-values of the last completed movements of all players for each episode. In our experiments, we define the duration of movements to be 3 seconds, so that the last completed action starts 3 seconds before the end of an episode. The computed $Q$-values are averaged over all agents to compute a single value for every episode which is considered a prediction of the outcome of an episode. Hence, the higher the average $Q$, the higher the probability that the attack is successful. We compute the area under the ROC curve (AUC) for each episode to compare predictions with actual rewards. We refer to this measure as *Q AUC*.

In order to evaluate $A$, we compute the advantage function (Equation 3) for every observed movement in an episode for all agents. The values are averaged over all agents and time steps to compute a single value for every episode. Analogous to *Q AUC*, these values are considered a prediction of the outcome of an episode, and are denoted as *Advantage AUC* in the results.

To give an intuition of choosing the evaluation measures, first recall that $Q(s,m^i)$ is the expected cumulative reward of $m^i$. Therefore, when we consider an agent's last movement in an episode, i.e. the intended movement 3 seconds before the end of the episode, its value reflects the expected outcome of the episode which serves as a suitable predictor of the outcome. On the other hand, advantage function $A$ describes the relative value of an action. The second evaluation measure thus evaluates that the more *qualified* decisions the agents make during an episode, the higher the likelihood of success would be.

**Performance Results:** Figure 5 (left, top) shows the average $Q$ AUC for varying numbers of mixture components of $\pi$, with a prediction horizon of 3 seconds. The AUC values of our model grow slightly with the complexity of the model and significantly outperform the baseline in predicting $Q$-values. This result strongly indicates that using our model-based approach proves advantageous when learning $Q$ from limited data.

Figure 5 (left, bottom) plots the average Advantage AUC which also increases with the complexity of the model. The third baseline, that uses the learned $V$, is not shown in the plot due to giving a value of 0.489 in our experiments. The *Baseline with Random Sampling* performs poorly, nonetheless, the *Baseline with Model* that uses the same movement model as our model performs comparable to our approach. This highlights the benefit of learning a movement policy in order to asses the

---

[1]We consider all movements with a speed below 30km/h possible.

[2]And in a real-world application, vice versa, situations in which agents frequently choose suboptimal actions could be reinforced in the coaching.

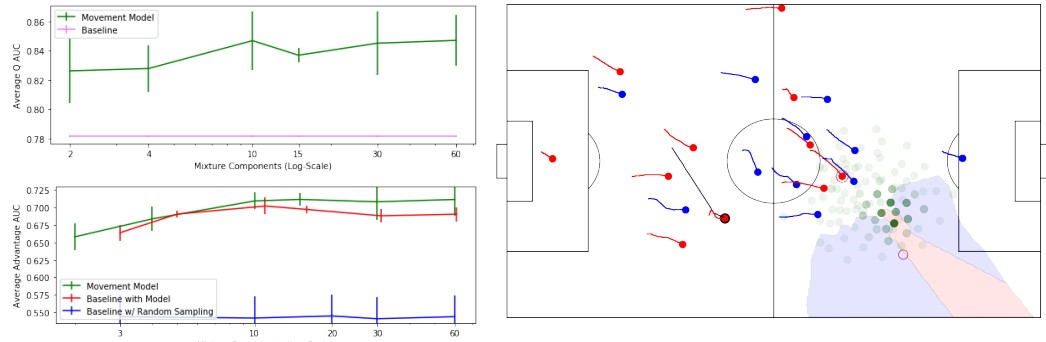

Figure 5: Top left: Average AUCs for $Q$-evaluation. Bottom left: Average Advantage AUCs. Right: Exemplary movement model prediction and occupation zones (3 seconds into the future).

relative value of movements, where our model allows to sample efficiently. Even though our model performs only slightly better than the informed baseline, both approaches significantly outperform the baselines without movement models,

In an additional evaluation, we compare our computed movement ratings to expert ratings. We average the advantage of the true movements of nine Dutch players (excluding the goalkeeper) who appeared in all six games over all timesteps in the episode, and compute their performance via cross validation. The models employ $k = 5$ mixture components and the movement duration is two seconds. We compare our ratings with expert rankings based on Space Occupation Gain (SOG) measure introduced by Fernandez & Bornn (2018) as well as scores from a newspaper[3]. Table 1 shows that our approach is in line with the two expert views.

Table 1: Players ratings

| SOG-based | | Our Model | | Newspaper | |
|---|---|---|---|---|---|
| 1:Sanden | 1.6 | 1:Sanden | 0.086 | 1:Martens | 8.5 |
| 2:Miedema | 1.5 | 2:Miedema | 0.032 | 2:Groenen | 8 |
| 3:Martens | 1.3 | 3:Donk | 0.005 | 3:Sanden | 7.5 |
| 4:Donk | 1.2 | 4:Groenen | 0.004 | 3:Miedema | 7.5 |
| 5:Groenen | 1.1 | 5:Martens | 0.003 | 4:Donk | 7 |
| 6:Spitse | 0.85 | 6:Dekker | 0.002 | 4:Dekker | 7 |
| 7:Van Es | 0.8 | 7:Gragt | -0.015 | 4:Gragt | 7 |
| 8:Dekker | 0.69 | 8:Spitse | -0.018 | 4:Spitse | 7 |
| 9:Gragt | 0.67 | 9:Van Es | -0.075 | 4:Van Es | 7 |

### 4.3 MOVEMENT MODEL

An interesting application of the learned movement policies is to compute the possible whereabouts and their likelihood for all players at $t$ seconds in the future. The player with the highest likelihood wins a position. The resulting area won by an agent is sometimes called an agents dominant region as she is likely to control this area (Taki & Hasegawa, 2000). Figure 5 (right) shows the same game situation as in Figure 1. The green dots visualize $k = 90$ mixture components of the player emphasized by the red circle; darker green corresponding to higher likelihood. Prediction is performed for $t = 3$ seconds into the future. The red zone denotes the controlled zone of that player and the two blue zones correspond to the two blue defenders below and right of the player. We show more examples and evaluations of the likelihood of the movement models in Appendix C.2.

### 5 CONCLUSION

We presented a purely data-driven approach to learning to rate agents' actions in collective spatial environments. On the example of soccer, we took an MDP-based approach where agents perceive positions and attributes of other players as the state representation and perform their movements from the continuous action space to pursue a joint goal. We proposed an efficient deep architecture to simultaneously optimize a policy which models realistic movements and a value function that rates them. The choice of model facilitates efficient learning and sampling of actions. Empirically, our model successfully captured the contribution of individual movements to the collective goal.

---

[3]https://www.ad.nl/nederlands-voetbal/rapport-glanzende-cijfers-voor-de-oranje-leeuwinnen a986670a0/

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

# Appendices

## A   PROOF OF PROPOSITION 1

The integral in Equation 4 can be written as a sum over the components of the mixture model

$$\int_{\tilde{m}^{+i}} \pi(\tilde{m}^{+i}|s^+)\bar{Q}(s^+, \tilde{m}^{+i})\, d\tilde{m}^{+i} = \sum_l \hat{\pi}(l|s^+, i)\hat{Q}(s^+, \hat{m}^l(s^+, i)). \tag{9}$$

*Proof.* We have

$$\int_{\tilde{m}^{+i}} \pi(\tilde{m}^{+i}|s^+)\bar{Q}(s^+, \tilde{m}^{+i})d\tilde{m}^{+i}$$

$$= \int_{\tilde{m}^{+i}} \pi(\tilde{m}^{+i}|s^+) \sum_l p(\hat{m}^l(s^+, i)|s^+, m^{+i})$$

$$\hat{Q}(s^+, \hat{m}^l(s^+, i))d\tilde{m}^{+i}$$

$$= \int_{\tilde{m}^{+i}} \sum_l p(\hat{m}^l(s^+, i), m^{+i}|s^+)\hat{Q}(s^+, \hat{m}^l(s^+, i))d\tilde{m}^{+i}$$

$$= \sum_l \int_{\tilde{m}^{+i}} \hat{\pi}(l|s^+, i)$$

$$\mathcal{N}(m^{+i}|\hat{m}^l(s^+, i), \sigma^l(s^+, i))\hat{Q}(s^+, \hat{m}^l(s^+, i))d\tilde{m}^{+i}$$

$$= \sum_l \hat{\pi}(l|s^+, i)\hat{Q}(s^+, \hat{m}^l(s^+, i))$$

where $\int_{\tilde{m}^{+i}} \mathcal{N}(m^{+i}|\hat{m}^l(s^+, i), \sigma^l(s^+, i))d\tilde{m}^{+i} = 1$. $\qquad\square$

Table 2: Per-layer definition of the convolutional model.

|  | kernel shape | #kernels | max. pooling |
|---|---|---|---|
| layer 1 | (5,5) | 32 | – |
| layer 2 | (5,5) | 64 | (2,2) |
| layer 3 | (5,5) | 64 | – |
| layer 4 | (5,5) | 128 | (2,2) |
| layer 5 | (5,5) | 128 | (2,2) |
| layer 6 | (5,5) | 128 | (2,2) |

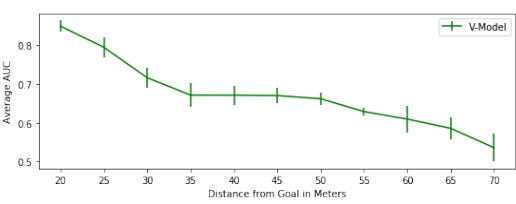

Figure 6: Average V AUC.

# B    DESCRIPTION OF THE DEEP MODEL

**Input:** The input to the network is represented by a three dimensional 'image-like' tensor $\mathbf{I} \in \mathbb{R}^{105 \times 68 \times 12}$ shown in Figure 2. The tensor encodes the size of the pitch ($105 \times 68$) and provides 12 channels that are used as follows: Channels 1-3 contain the positions of the agents (channels 1-2 for team red and team blue) and the ball (channel 3), respectively; tensor elements corresponding to the position of an agent or the ball are set to 1. Channels 4-9 contain the directions of movements and velocities of agents of team red (channels 4-5), team blue (channels (6-7)), and the ball (channels 8-9). By doing so, input features that belong to a specific agent are given by the slice $(x, y, :)$ that is located at the agent's $(x, y)$ position. Further channels encode ball possession (channel 10), the position of the goals (channel 11) as well as the boundaries of the pitch (channel 12). Note that channels 11 and 12 are constant and do not change over time.

**Architecture:** The detailed definition of layers in deep convolutional net is sketched in Table 2.

**Parameters:** Recall that the feature representation of the agents serves as the input to the fully connected layer generating all outputs (Figure 4). Let $\Psi_{conv}$ be the set of all parameters of the convolutional net except for the fully connected layer. Furthermore, let $\Psi_{\hat{Q},O} \in \mathbb{R}^{d,k}$ denote the parameters of the fully connected layer that are involved in the computation of $\hat{Q}$. Analogously, $\Psi_{\hat{\pi},O} \in \mathbb{R}^{d,k}$, $\Psi_{\sigma,O} \in \mathbb{R}^{d,k}$, and $\Psi_{\hat{m},O} \in \mathbb{R}^{d,2k}$ denote the parameters for $\hat{\pi}$, $\sigma$, and $\hat{m}$, respectively. Hence, the layout of the net imposes a massively shared structure between the convolutional layers and the quantities of interest. In other words, the parameters of the convolutional net $\Psi_{conv}$ are shared between all agents and outputs and we have $\Psi_\omega = \Psi_{conv} \cup \Psi_{\omega,O}$, for all $\omega \in \{\hat{Q}, \hat{\pi}, \hat{m}^l, \hat{\sigma}^l\}$.

**Optimization:** For a given input representation of a state $s$, the deep model outputs $\hat{\pi}(l|s,i)$, $\hat{m}^l(s,i)$, $\sigma^l(s,i)$, and $\hat{Q}(s, \hat{m}^l(s,i))$ for all agents $1 \le i \le n$ and mixture components $1 \le l \le k$. The parameters of the model are adjusted by minimizing the compositive objective function,

$$\min_\Psi \quad (\Omega_{\bar{Q}}(\Psi_{\hat{Q}}; \Psi_{\hat{\pi}}, \Psi_{\hat{m}}, \Psi_{\hat{\sigma}}) + \Omega_L(\Psi_{\hat{\pi}}, \Psi_{\hat{m}}, \Psi_{\hat{\sigma}})),$$

with standard optimizers for deep models such as stochastic gradient descent.

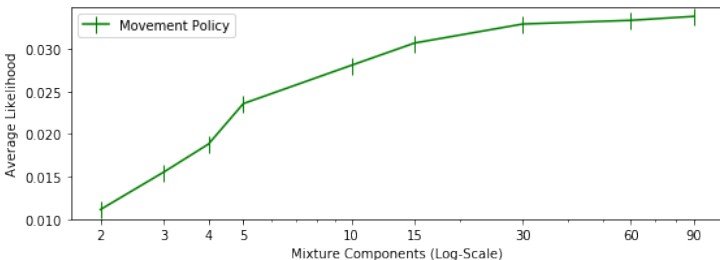

Figure 7: Likelihood for different numbers of components in the movement mixture.

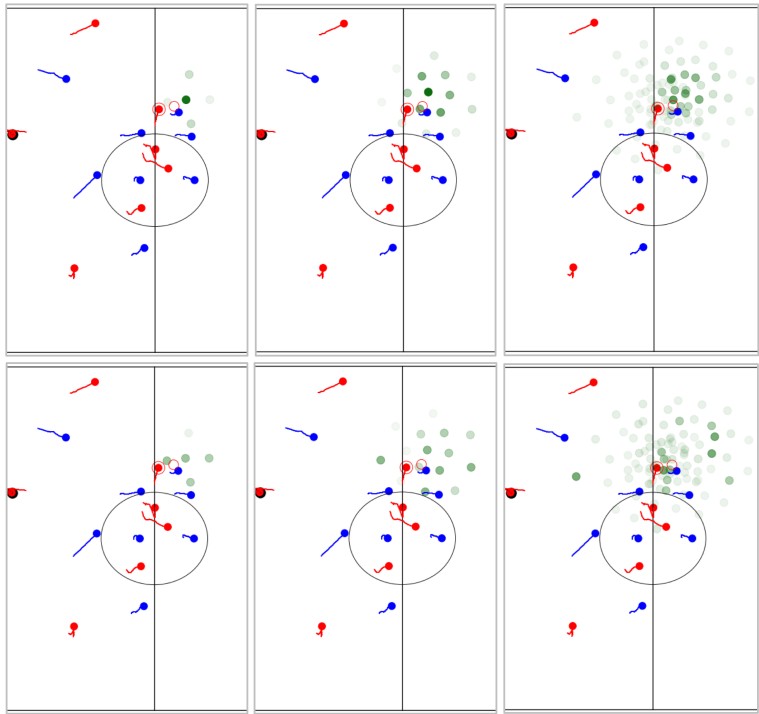

Figure 8: Predicted movements (top row) and advantages (bottom row) with mixtures using $k = 5$ (left), $k = 15$ (center), and $k = 90$ (right). Darker green marks higher likelihood (top row) and higher advantage values (bottom row), respectively. Predictions are shown for the circled player and the real next position is depicted by an empty circle.

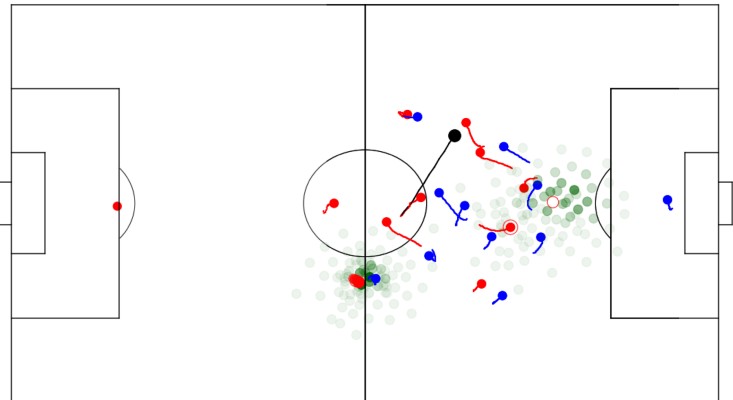

Figure 9: The slowly moving agent on the left has a circular-shaped movement distribution as there is only little time needed for turning in any direction. The running agent on the right (marked by the red circle) realizes an ellipsoidal-shaped movement distribution.

## C    ADDITIONAL EVALUATIONS

### C.1    $V$ EVALUATION

To get a better understanding of the quality of the estimated $V$ of the third baseline, figure 6 shows the performance of state valuation in terms of average AUCs. We compute the value of a frame in a positively labeled episode and compare it with all frames of all negatively labeled episodes where the ball has the same distance to the opponent goal. We hence estimate the probability that $V$ assigns

higher scores to a frame from a successful episode than to one of a negative episode. Unsurprisingly, the closer the ball is to the opponent's dangerous zone, the better the AUC.

## C.2 MOVEMENT MODEL

The movement policy is learned by adapting a Gaussian mixture model to the data points. Intuitively, the more mixture components, the better the approximation of the continuous space. Figure 7 confirms this and shows that the likelihood increases for further numbers of components, where the model predicts movements that span 3 seconds. See also Figure 8 (top row) for an exemplary situation with $k = \{5, 15, 90\}$ components. The bottom row demonstrates the advantage values of the respective movements.

Furthermore, the movement policy nicely adapts to the context of the player as shown in Figure 9 which depicts two different exemplary situations of predicted movements by a learned policy with $k = 90$ mixture components. The running agent on the right realizes a very different movement distribution than the slowly moving player on the left.

