# OpenReview forum: "Rating Continuous Actions in Spatial Multi-Agent Problems"
_ICLR.cc/2019/Conference_

### Official Review · AnonReviewer2 · 2018-10-29
**Somewhat interesting application, unclear technical contribution.**

**Rating:** 4
**Confidence:** 4

**Review:**

This paper presents a method for learning the Q function from multi-agent demonstrations, such as trajectories of soccer players in a soccer game.  The basic ideas are:

-- The policy class pi(m|s) maps states s to actions m probabilistically.  In particular, the probability class is a mixture of Gaussians, with the mean/covariance functions and the mixture function all being instantiated via neural nets.  The behavior of the individual players are assumed to be conditionally independent given the current state.

-- A hand-designed reward function is used, such as the ball being in a "strike zone" of some kind for the team on offense.

-- The Q function class is also a neural net.

-- The policy pi is learned through maximum likelihood of the state/action pairs of the the pre-collected demonstrations.  This is essentially probabilistic behavioral cloning over the policy class.

-- The Q function is learned via Q-fitting over the demonstration data, i.e., minimizing squared error on the Bellman residual.  This requires the pre-specified reward function, the policy pi, and some details regarding the state transitions (e.g., the non-deterministic transitions of the ball).

-- The policy pi and the Q function are trained jointly by adding the two objectives.

-- The main empirical evaluation is comparing the quality of the Q function estimation against a naive baseline (random movements) and a non-mixture policy class (i.e., a single mode, rather than a mixture of Gaussians).  The results against the non-mixture policy class do not seem to show much difference in performance.

-- Some qualitative evaluations are also presented, but it's unclear what insight one is expected to gain from looking at them.


**Clarity**
The paper is reasonably well written.  Some aspects of the logical flow could be improved, but overall it's fine.  Detailed comments are:

-- The proposed approach is essentially imitation learning and not, strictly speaking, reinforcement learning.  Yet the work is positioned as reinforcement learning.  Perhaps it's just a difference in norms of terminology usage.

--  There are more multi-agent RL & IL works than what was discussed in the related works section.  Of course, an exhaustive discussion is not expected, but the writing makes it sound like there hasn't been much work at all.  Examples include:
https://arxiv.org/abs/1807.09936
https://arxiv.org/abs/1706.02275
http://www.cs.ox.ac.uk/people/shimon.whiteson/pubs/rashidicml18.pdf

-- The authors comment in the related work that team sports cannot yet be simulated.  This statement is sort of true, although it's not clear how true. For instance, the physics engines of many sports video games are very realistic.  Moreover, it doesn't strike me as the most prominent point of contrast with [Hausknecht & Stone 2016].  For instance, a more prominent point of contrast is simply imitation learning vs reinforcement learning.

-- It's not clear what the authors mean specifically when they refer to "average performance" contrasting with [Le et al. 2017b].  Also, desn't that work also condition on game context and situation?

-- Section 4.1 is written in a somewhat disorganized way.  The exposition alternates between discussing baselines and the evaluation methodology in a way that is confusing.

-- One of the baselines is described but then left to the appendix for analysis.  This doesn't make sense from a narrative perspective.


**Originality**
From a technical perspective, it's not clear that there's much novelty in this approach.  All the ingredients are pretty standard, and the ingredients are put together in a standard way.

From an applications perspective, there might be some novelty here but it's not clear.  The application of imitation learning to sports data is not new, but also not saturated either.  The specific idea of learning a better rating system is interesting, but it's not clear how fleshed out this application is in the paper (more comments on this below).


**Significance**
For me, the key question for significance boils down to: "Does this paper change the way people think about doing X?" for the most interesting instantiation of X possible.  In this case, it seems X should be computing a rating system for multi-agent spatial multi-agent behavior (as suggested by the title).  But the limitations in the model and the experiments make this a questionable proposition.

The model class all but ignores the multi-agent aspect (the agents are all independent conditioned on the states). Moreover there was no comparison against a model class that made a worse assumption on how to handle the multi-agent aspect.  So it's not clear how the multi-agent aspect is significant.

The evaluation of the rating approach is not convincing.  The approach essentially performed as well as a single-mixture baseline.  Furthermore, the direction V(s) learning approach is deferred to the appendix, and I'd like to see it included in the main paper.  Finally, it would be nice if there was some attempt to compare with a non-MDP based approach.  For instance, people directly compute things like P(goal|state).  I'd like to see evidence that this style of approach will outperform those approaches, which are the de facto standard in sports analytics right now.


**Overall Quality**
Based on the above comments, it is my opinion that this paper has too many holes in it to be ready for publication at a venue such as ICLR.  As a primarily applications paper, the burden is on the authors to demonstrate reasonably convincing evidence that this line of approach is valuable to the application.

---

### Official Review · AnonReviewer1 · 2018-10-31
**a bit too specific in the application, limited results and comparisons of the method with SOTA**

**Rating:** 4
**Confidence:** 3

**Review:**

The paper proposes to learn a policy and action value over continuous next movement from professional soccer game. The policy is modeled as a gmm and the value of the mean of each component is learned jointly.

The experiment section seems a bit light. The dataset seems quite small, would be good to get learning curves on training / test sets separately to see of the model is overfitting. Also a breakdown of results with bigger / deeper convnets rather than just number of mixture components would be interesting. There is not much comparison with other ML / RL types methods.

p3: why not ' instead of + to denote next state ?

f+i = fi + mi : why no f^{i+1}

Nevertheless, the movement of
the ball is a part of the environment and is modeled as random movement. Therefore, the movement
distribution is governed by a stochastic policy.

Even if the ball movement was deterministic, the movement could still be stochastic.

Eq (2) right hand side, inconsistency between p(m | s, m) with p(s+ | s, m) defined above ?

Eq 6: I guess a softmax over a grid centered around player position would also work ? Possibly convolutional all the way? It wouldnt have much more parameters than the proposed arch?

---

### Official Review · AnonReviewer3 · 2018-11-02
**Weak evaluation**

**Rating:** 5
**Confidence:** 4

**Review:**

The paper studies the credit assignment problem in multi-agent domain (soccer playing as a concrete application). The paper itself is well-written. I hope that the author could use a better methodology to make the evaluation part stronger.

I have a few questions to the author:

1. Why features like which team has ball possession (a boolean) is represented as a channel in the image? What if you concatenate this bit with the output of the last layer of convolution?

2. Compared to the proposed algorithm, the evaluation part is somewhat weaker.
 i) I understand that it might be hard to get ground truth of the credit assignment, but is there any way to get a more competitive baseline model.
ii) the dataset you used only contains 6 games from the same team in one tournament.  This seems biased and not a reliable dataset itself.

Minor typos:
* section 4. We extract all episodes of open play (where) one team

---

### Meta-Review · Area_Chair1 · 2018-12-20

**Confidence:** 5
**Recommendation:** Reject

**Metareview:**

The reviewers raised a number of major concerns including the incremental novelty of the proposed (if any), a poor readability of the presented materials, and, most importantly, insufficient and unconvincing experimental evaluation presented. The authors did not provide any rebuttal. Hence, I cannot suggest this paper for presentation at ICLR.